# Activation of RAAS Signaling Contributes to Hypertension in Aged *Hyp* Mice

**DOI:** 10.3390/biomedicines10071691

**Published:** 2022-07-13

**Authors:** Nejla Latic, Ana Zupcic, Danny Frauenstein, Reinhold G. Erben

**Affiliations:** Department of Biomedical Sciences, University of Veterinary Medicine, Veterinaerplatz 1, 1210 Vienna, Austria; nejla.latic@vetmeduni.ac.at (N.L.); ana.zupcic@vetmeduni.ac.at (A.Z.); danny.frauenstein@gmail.com (D.F.)

**Keywords:** fibroblast growth factor-23, left ventricular hypertrophy, hypertension, renin–angiotensin–aldosterone system, losartan, canrenone, *Hyp* mice, X-linked hypophosphatemia

## Abstract

High circulating levels of fibroblast growth factor-23 (FGF23) are associated with left ventricular hypertrophy as well as increased morbidity and mortality in patients suffering from chronic kidney disease. However, the mechanisms underlying this association are controversial. Here, we aimed to further characterize the cardiovascular sequelae of long term endogenous FGF23 hypersecretion using 14-month-old male *Hyp* mice as a model of FGF23 excess. *Hyp* mice were characterized by a ~10-fold increase in circulating intact FGF23, hypophosphatemia, increased serum aldosterone, but normal kidney function, relative to wildtype (WT) controls. Cardiovascular phenotyping did not reveal any evidence of left ventricular hypertrophy or functional impairment in 14-month-old *Hyp* mice. Fractional shortening, ejection fraction, molecular markers of hypertrophy (*Anp*, *Bnp*), and intracardiac markers of contractility and diastolic function were all unchanged in these animals. However, intraarterial catheterization revealed an increase in systolic, diastolic, and mean arterial pressure of ~12 mm Hg in aged *Hyp* mice relative to WT controls. Hypertension in *Hyp* mice was associated with increased peripheral vascular resistance. To test the hypothesis that a stimulation of the renin–angiotensin–aldosterone system (RAAS) contributes to hypertension in aged *Hyp* mice, we administered the angiotensin receptor blocker losartan (30 mg/kg twice daily) or the mineralocorticoid receptor antagonist canrenone (30 mg/kg once daily) to aged *Hyp* and WT mice over 5 days. Both drugs had minor effects on blood pressure in WT mice, but reduced blood pressure and peripheral vascular resistance in *Hyp* mice, suggesting that a stimulation of the RAAS contributes to hypertension in aged *Hyp* mice.

## 1. Introduction

The most frequent inherited renal phosphate-wasting disease in humans is X-linked hypophosphatemia (XLH). The murine homolog is *Hyp* (hypophosphatemia). XLH patients and *Hyp* mice lose phosphate via the urine and show impaired bone mineralization as a consequence of hypophosphatemia and alterations in the organic bone matrix. The disease is caused by loss-of-function mutations in *PHEX* (Phosphate-regulating endopeptidase homolog, X-linked) in mice and men. By mechanisms that are poorly understood, *PHEX* mutations lead to excessive secretion of the bone-derived, phosphotropic hormone fibroblast growth factor-23 (FGF23) [1,2,3].

FGF23 is a 32 kDa glycoprotein circulating in the bloodstream. Only the intact molecule is biologically active. FGF23 signals through FGF receptors (FGFRs), with FGFR1c being the most important receptor under physiological conditions. High affinity binding of FGF23 to FGFRs in the cell membrane requires the presence of the co-receptor protein αKlotho [4,5]. The major site of action of FGF23 is the kidney, where it inhibits tubular reabsorption of phosphate and suppresses vitamin D hormone (1,25(OH)_2_D) production in proximal renal tubules [6]. Excessive concentrations of intact FGF23 lead to renal phosphate wasting, hypophosphatemia, and impaired bone mineralization in patients with normal kidney function [7].

Although FGF23 mainly acts on epithelial cells in the kidney under physiological conditions, other organs such as the cardiovascular system may become targets of FGF23 signaling in pathological settings. The cardiovascular sequelae of chronically elevated FGF23 are still controversial. Studies in chronic kidney disease (CKD) patients have shown that elevated FGF23 levels are independently associated with CKD progression, left ventricular (LV) hypertrophy, cardiovascular risk, and all-cause mortality [8,9,10]. A key question in this context is whether FGF23 is only a biomarker of impaired phosphate metabolism or chronic inflammation or whether there is a causal link between FGF23 and increased cardiovascular risk. The myocardium does not express αKlotho [10,11], so the heart is likely not a target of FGF23 action under normal conditions. However, at high circulating concentrations such as those found in CKD patients, FGF23 may promote LV hypertrophy by a direct, αKlotho-independent, FGFR4-mediated action on cardiomyocytes [10,11,12]. Alternatively, FGF23 excess may cause LV hypertrophy by contributing to volume overload through its sodium-conserving effect [13].

Whether LV hypertrophy is typically associated with other diseases (than CKD) characterized by FGF23 excess, such as XLH, is unclear. Some studies find a high incidence of LV hypertrophy in XLH patients [14,15], but others report no such association [16]. In contrast, the data from experimental studies are more clear-cut: several independent studies failed to find LV hypertrophy in *Hyp* mice [17,18] or in mice with a loss-of-function point mutation in the *Phex* gene [19]. Although LV hypertrophy appears to be absent at least in younger *Hyp* mice up to 9 months of age, we and others found mild hypertension in these mice [6,13]. The pathophysiological mechanisms underlying hypertension in *Hyp* mice are not entirely clear, but they may be related to an FGF23-mediated upregulation in renal sodium-chloride cotransporter (NCC) abundance [13]. Injection of recombinant FGF23 for five days into wild-type mice led to an increase in systolic, diastolic, and mean arterial pressure [13]. *Hyp* mice are characterized by chronic elevation of FGF23 and may, therefore, serve as a model to investigate the pathophysiology of hypertension in diseases associated with excessive FGF23 secretion.

An improved understanding of the mechanistic link between chronically elevated FGF23 and LV hypertrophy or other untoward cardiovascular side effects such as hypertension is of major importance, not only for XLH patients but for all diseases characterized by chronic elevations of circulating intact FGF23, such as CKD. Here, we sought to further elucidate the long term cardiovascular sequelae of elevated circulating levels of FGF23 in 14-month-old *Hyp* mice as a model of FGF23 excess. We found that aged *Hyp* mice lacked LV hypertrophy but were characterized by increased serum aldosterone and hypertension that could be rescued by inhibitors of the renin–angiotensin–aldosterone system (RAAS).

## 2. Materials and Methods

### 2.1. Animals

The study was undertaken in accordance with prevailing EU and national guidelines for animal care and welfare and in compliance with ARRIVE (Animal Research: Reporting of In Vivo Experiments) guidelines. All animal procedures were approved by the Ethics and Animal Welfare Committee of the University of Veterinary Medicine Vienna, Austria, and by the Austrian Federal Ministry of Education, Science and Research (permit number BMWFW-68.205/0054-II/3b/2013 and 2021-0.331.140).

Male wild-type (WT) controls and *Hyp* mice were bred by mating WT females with *Hyp* males on C57BL/6 background in our animal facility. Tail length at the time of weaning was used for genotyping. Animals were kept at 24 °C with a 12/12-h light/dark cycle, and were housed in stable groups of 2–5 mice from the time of weaning. They were fed a normal mouse diet (V1124-000, Sniff, Soest, Germany) containing 1.0% calcium, 0.7% phosphorus, and 1000 IU vitamin D/kg, and had access to food and tap water ad libitum. At necropsy, the mice were exsanguinated from the abdominal vena cava under general anesthesia with ketamine/medetomidine (100/0.25 mg/kg i.p.) for serum collection. No mice were excluded from the study.

### 2.2. Losartan and Canrenone Treatment

Some *Hyp* and WT mice were randomized to treatment with losartan (Sandoz, Austria, 50 mg) or vehicle (0.9% NaCl) via oral gavage twice a day or with subcutaneous injections of canrenone (Aldactone, Reimser, Germany, 200 mg/10 mL) or vehicle (0.9% NaCl) daily. Both treatments were given for 5 days.

### 2.3. Biochemical Analysis

Serum sodium, phosphate, and calcium were measured using a Cobas c111 analyzer (Roche, Mannheim, Germany). Serum intact FGF23 (Kainos, Tokyo, Japan), aldosterone (NovaTec Immundiagnostica, Dietzenbach, Germany), and intact PTH (Immutopics Inc., San Clemente, CA, USA) were detected using commercially available ELISA kits according to the manufacturer’s instructions. Serum was extracted with diethylether and re-suspended in steroid-free serum (DRG Diagnostics, Marburg, Germany) for the aldosterone ELISA.

### 2.4. Echocardiography

Echocardiography was performed one day before the necropsies using a 14 MHz linear transducer (Siemens Accuson s2000, Munich, Germany) for evaluation of cardiac function. Mice were under 1% isoflurane anesthesia with a stable body temperature of 37 °C. M-mode in short-axis at the level of the papillary muscles was used to evaluate LV thickness, fractional shortening, and internal diameters in systole and diastole. At least four cardiac cycles were analyzed for each parameter.

### 2.5. Central Arterial and Cardiac Pressure Measurements

Central arterial pressure was measured by inserting a micro-tip catheter (1.4 Millar Instruments, Houston, TX, USA) into the ascending aorta via the right carotid artery under 1.5% isoflurane anesthesia. The catheter was then further advanced into the left ventricle to obtain cardiac pressure parameters. Traces were recorded for at least three minutes and analyzed via LabChart7 software (ADInstruments, Dunedin, New Zealand).

### 2.6. Augmentation Index

The aortic augmentation index was identified from the late systolic portion of the arterial pressure wave as described previously [20]. The augmentation index was defined as the height from the augmentation point to the systolic peak of the pressure wave divided by the pulse pressure and was expressed as a percentage.

### 2.7. Western Blot

Fresh frozen kidneys were homogenized in RIPA lysis buffer supplemented with phosphatase inhibitor cocktail (Roche) and protease inhibitor cocktail (Roche). Kidney tissue homogenates were mixed with Laemmli sample buffer, fractionated on SDS-PAGE (20 μg/well), and transferred to a nitrocellulose membrane (Thermo Fisher Scientific, Waltham, Massachusetts, USA). Immunoblots were incubated overnight at 4 °C with primary antibodies, including monoclonal rat anti-human Klotho (1:500, KO603, Trans Genic Inc., Tokyo, Japan), polyclonal rabbit anti-phospho-NCC (pThr53, 1:1000, Novus Biologicals, Littleton, CO, USA), and monoclonal mouse anti-β-actin (1:5000, Sigma, St. Louis, MS, USA) in 2% (*w*/*v*) bovine serum albumin (BSA, Sigma) and washed in TBS-T buffer (150 mM NaCl, 10 mM Tris/HCl pH 7.4, 0.2% *v*/*v* Tween-20). After washing, membranes were incubated with horseradish peroxidase-conjugated secondary antibodies (Amersham Biosciences, Buckinghamshire, UK). Specific signal was visualized using ECL kit (Amersham Life Sciences, Arlington Heights, IL, USA). The protein bands were quantified by Image Studio Lite 5.2 software (LI-COR, Bad Homburg, Germany).

### 2.8. Histological Evaluation

Hearts were fixed in 4% paraformaldehyde, paraffin-embedded, and cut into 5 μm sections. Fibrotic tissue was visualized by staining with picrosirius red according to a standard protocol. Total collagen was quantified using ImageJ software and was expressed as the ratio of collagen-stained area to total muscle area of the left ventricle and septum. For the analysis of cardiomyocyte size, cardiac sections were stained with FITC-labeled wheat germ agglutinin. At least ten random areas of the heart were measured, and only cardiomyocytes with well-defined borders and visible nuclei were used. Images were obtained by the Zeiss LSM 880 Airyscan confocal microscope and analyzed using semi-automated Image J software. All histological images were analyzed by two independent investigators in a blinded manner.

### 2.9. Immunofluorescence

Kidneys fixed in paraformaldehyde (PFA)-fixed were cut in 5-μm-thick paraffin sections. Sections were dewaxed, demasked for 20 min with proteinase K, and, after washing, pretreated with 10% normal goat serum in PBST for 60 min at room temperature (RT). Without rinsing, sections were incubated with polyclonal rabbit anti-pNCC (Novus Biologicals, 1:200) antibody at 4 °C overnight. After washing, sections were incubated for 1 h with goat anti-rabbit Alexa 594 (1:500, Invitrogen, Waltham, MA, USA). Nuclear staining was performed with DAPI (1:1000) for 5 min. Controls were performed by omitting primary antibodies. The slides were analyzed on a Zeiss LSM 880 Airyscan confocal microscope equipped with a 63× oil immersion lens (NA 1.3).

### 2.10. RNA Isolation and Quantitative RT-PCR

Total RNA was isolated from snap-frozen hearts after homogenization in a 24 Fast Prep machine using TRI Reagent. The nucleic acid concentration and integrity were determined by electrophoresis (Agilent Tapestation). Only samples that had a RIN value above seven were used. Two micrograms of RNA were transcribed into cDNA using the High Capacity cDNA Reverse Transcription Kit (Applied Biosystems, Waltham, MA, USA). Quantitative RT-PCR was performed on a qTOWER3/G qPCR device (Analytic Jena, Jena, Germany). The qPCR performed in a volume of 15 μL was composed of 1× PCR buffer B2 (Tris-HCl, (NH_4_)_2_SO_4_, and Tween-20; Solis Biodyne, Tartu, Estonia), 3.5 mM MgCl_2_, 200 μM dNTP mix (Solis Biodyne), 250 nM of each primer, and either 200 nM hydrolysis probe or 0.4 × EvaGreen (Biotium, Fremont, CA, USA), 1 U HOT FIREPol^®^ DNA polymerase (Solis Biodyne), and 2 μL template DNA. For mouse primer sequences, see Appendix A. Cycling conditions consisted of an initial 15-min incubation step at 95 °C for polymerase activation and template denaturation, followed by 45 cycles of 95 °C denaturation for 15 s and 60 °C annealing and elongation for 60 s. All samples were measured in triplicate and normalized to two housekeeping genes (*Dpm1* and *Txnl4a*). The qPCR results were gained and primarily evaluated with the software qPCRsoft 4.1 (V4.1.3.0 Analytic Jena) and then analyzed using the standard delta delta Cq method.

### 2.11. Statistical Analysis

Statistical analysis was performed using Graph Pad Prism 9 or SPSS. The data were analyzed by two-sided t-test for two groups, one-way or two-way analysis of variance (ANOVA) to assess the influence of genotype and treatment, as well as their two-way interactions followed by the Student–Newman–Keuls multiple comparison test when comparing more than two groups; *p*-values of less than 0.05 were considered significant. Data are presented as scatter dot plots with bars depicting means ± SEM.

## 3. Results

Aged *Hyp* mice show a stimulation of the RAAS and a downregulation of renal αKlotho protein abundance, but they have normal kidney function.

As expected, 14-month-old male *Hyp* mice were characterized by an approximately 10-fold increase in serum intact FGF23 when compared to age- and sex-matched WT control animals (Figure 1A). The elevated circulating intact FGF23 levels in aged *Hyp* mice were associated with hypophosphatemia (Figure 1B). Serum calcium tended to be lower, whereas serum intact PTH tended to be higher in *Hyp* mice, but both effects did not reach statistical significance (Figure 1C,D). We reported earlier that 3-month-old *Hyp* mice were characterized by lower serum aldosterone levels relative to WT mice [13]. However, in aged *Hyp* mice, serum aldosterone was not down-, but rather up-regulated (Figure 1E). Aldosterone and FGF23 are known regulators of the sodium chloride cotransporter NCC activity in distal convoluted renal tubules [13]. Therefore, we quantified renal pNCC protein abundance by Western blotting and by immunofluorescence analysis. Similar to younger *Hyp* mice [13], renal pNCC levels were up-regulated and pNCC immunofluorescence tended to be higher in 14-month-old *Hyp* mice, relative to WT controls (Figure 1F,G). The abundance of αKlotho protein was distinctly down-regulated by about 50% in the kidneys of *Hyp* mice (Figure 1H), which may be a counter-regulatory mechanism to protect against the chronically elevated circulating FGF23 levels. However, renal function as evidenced by serum creatinine concentration and by renal creatinine clearance remained unchanged in aged *Hyp* mice (Figure 1I,J).

### 3.1. Elevated Levels of Circulating FGF23 Are Associated with Mild Hypertension in Aged Hyp Mice but Do Not Cause LVH

Several earlier reports have failed to find LV hypertrophy in *Hyp* mice of up to nine months of age [17,18]. However, data about potential changes in cardiovascular function in aged *Hyp* mice as a model of long term FGF23 excess are scarce. Therefore, we examined the cardiovascular phenotype of aged *Hyp* mice by intraarterial and left ventricular catheterization as well as by echocardiography. The HW/BW ratio was distinctly increased in *Hyp* mice relative to WT controls (Figure 2A). However, this increase was mainly driven by lower body weight in *Hyp* mice, and not by a higher heart mass (Figure 2B,C), questioning the relevance of the HW/BW ratio as a read-out for heart hypertrophy in this animal model. Notably, intraarterial catheterization revealed a higher mean arterial pressure (MAP) of about 12 mmHg in aged *Hyp* mice compared to WT controls (Figure 2D). Despite the presence of hypertension, LV function as evidenced by fractional shortening and ejection fraction measured by echocardiography was actually improved in *Hyp* compared with WT mice (Figure 2E,F). In agreement with the echocardiography data, LV contractility as measured by Max dP/dt during LV catheterization was unchanged in 14-month-old *Hyp* mice (Figure 2G). Moreover, there was no difference in mean cardiomyocyte size as measured by wheat germ agglutinin (WGA) staining or in LV collagen content as measured by picrosirius red staining between WT and *Hyp* mice (Figure 2H,I). The absence of LV hypertrophy in *Hyp* mice was further confirmed by qRT-PCR analysis of typical markers of hypertrophy, such as *Anp* and *Bnp,* which remained unaltered between the genotypes (Figure 2J,K). Taken together, these data confirm the presence of mild hypertension in aged *Hyp* mice but strongly argue against LV hypertrophy and functional impairment in these mice.

To further address the question of what drives hypertension in aged *Hyp* mice, we measured the augmentation index in the carotid artery by pulse wave analysis. The latter analysis revealed an increase in augmentation index in aged *Hyp* mice relative to WT controls (Figure 2L). This finding may point to increased peripheral vascular resistance in aged *Hyp* mice. In combination with the increase in aldosterone levels found in *Hyp* mice (Figure 1E), we hypothesized that hypertension in aged *Hyp* mice may be due to crosstalk between FGF23 and RAAS signaling in the kidney and blood vessels, leading to a combination of volume overload through increased aldosterone and FGF23 secretion and increased peripheral vascular resistance by elevated angiotensin II levels.

### 3.2. Inhibition of RAAS Signaling Normalizes Blood Pressure in Hyp Mice

To test this hypothesis, we administered the aldosterone antagonist canrenone and the angiotensin receptor blocker losartan over five days to 12- to 14-month-old male *Hyp* and WT mice. Intraarterial catheterization confirmed hypertension in vehicle-treated *Hyp* mice compared to WT controls. Interestingly, daily subcutaneous injections of 30 mg/kg canrenone led to a distinct decrease in systolic, diastolic, and mean arterial pressure in *Hyp* mice relative to vehicle-treated *Hyp* mice (Figure 3A–C). In contrast, WT mice injected with canrenone showed only minor reductions in mean, systolic, and diastolic blood pressure (Figure 3A–C). Losartan was administered via oral gavage at the dose of 30 mg/kg twice daily, and the effects of losartan were examined one hour after the last administration, due to a shorter half-life of this drug compared with canrenone. Similar to canrenone, losartan caused a pronounced decrease in systolic, diastolic, and mean arterial pressure in *Hyp* mice, relative to vehicle-treated *Hyp* mice, but had only negligible effects in WT animals (Figure 3D–F).

To determine whether the beneficial effect of RAAS inhibition on hypertension was associated with a decrease in peripheral vascular resistance, we analyzed the augmentation index in WT and *Hyp* mice post-treatment. Interestingly, both treatments reduced the elevated augmentation index in *Hyp* mice. Canrenone did not alter the augmentation index in WT mice, but significantly lowered the augmentation index in *Hyp* mice, whereas losartan had a similar effect in both genotypes (Figure 3G,H). Two-way ANOVA revealed a significant two-way interaction between genotype and treatment for systolic pressure (*p* < 0.001), mean arterial pressure (*p* < 0.01), and augmentation index (*p* < 0.05) in animals treated with canrenone, whereas this interaction was only found significant for the reduction in systolic pressure (*p* < 0.01) in *Hyp* mice treated with losartan (Figure 3A–H). Collectively, our data suggest that the blood pressure-lowering response of *Hyp* mice to both canrenone and losartan is exaggerated compared with WT mice, supporting our hypothesis that activation of RAAS signaling contributes to hypertension in aged *Hyp* mice. Because both canrenone and losartan reduced blood pressure and augmentation index in aged *Hyp* mice, it is likely that the contributing effect of the RAAS to the development of hypertension is mediated through a combination of increased peripheral resistance together with increased blood volume.

## 4. Discussion

The central aim of this study was to characterize the pathophysiological role of long term endogenous FGF23 hypersecretion in the cardiovascular system of 14-month-old male *Hyp* mice. We found that aged *Hyp* mice were mildly hypertensive but did not develop LV hypertrophy. Hypertension in aged *Hyp* mice was associated with increases in serum aldosterone levels, in vascular peripheral resistance, and in renal pNCC abundance. Administration of the angiotensin II receptor blocker losartan and the mineralocorticoid receptor blocker canrenone rescued the cardiovascular phenotype by lowering blood pressure and vascular resistance in *Hyp* mice.

Elevated circulating FGF23 levels are associated with accelerated disease progression, morbidity, and/or mortality in several clinical disorders, including CKD but also cardiac failure [8,9,10,21]. It was proposed that excessive FGF23 causes LV hypertrophy by Klotho-independent binding to FGFR4 and subsequent activation of the calcineurin/NFAT pathway in cardiomyocytes [11,12]. However, it is still unclear whether this disease mechanism is relevant in the setting of normal kidney function, because evidence from clinical studies in XLH patients and mouse models of XLH suggests that chronic FGF23 excess does not invariably cause LV hypertrophy [15,16,17,18]. In our study, an increase in circulating intact FGF23 levels of ~10-fold was not sufficient to induce an increase in LV size or deterioration of LV function in 14-month-old *Hyp* mice when compared to WT littermates. Fractional shortening, ejection fraction, and molecular markers of hypertrophy (*Anp*, *Bnp*) all remained unchanged in these animals. This is in line with previous studies in animal models of XLH that did not find any association between increased levels of FGF23 and LV hypertrophy [17,19].

The current study has shown that aged *Hyp* mice are characterized by a small increase in systolic, diastolic, and mean arterial blood pressure, corroborating earlier studies in younger *Hyp* mice [13,22]. Although hypertension is not a universal complication in XLH patients [14], early-onset hypertension is frequently found in adult XLH patients [23]. Based on the sodium-conserving function of FGF23, elevated circulating FGF23 may predispose subjects to the development of hypertension through volume expansion [13]. However, an increase in serum intact FGF23 levels of about 6-fold did not increase blood pressure in mice with a loss-of-function point mutation in the *Phex* gene that are characterized by normal kidney function [19]. Rather, the latter study reported that systolic pressure was actually slightly reduced in aged *Phex^C73^**^3R^* male mice relative to WT controls. A possible explanation for these discrepant findings may be the greater increase in FGF23 serum levels in *Hyp* mice. However, elevated circulating FGF23 levels per se may not be sufficient to cause hypertensive changes in the absence of other contributing factors.

Our data have revealed for the first time that serum aldosterone as well as augmentation index were increased in aged *Hyp* mice. Although we did not measure renin activity and angiotensin II levels, these findings suggest a general stimulation of the RAAS in aged *Hyp* mice. In contrast, we found lower aldosterone levels in 3-month-old *Hyp* mice [13]. We do not have a conclusive explanation for the discrepancy between young and aged *Hyp* mice regarding aldosterone secretion. There may be age-related changes in aldosterone secretion in *Hyp* mice or differences in the interaction between FGF23 signaling and the RAAS in young, growing vs. aged, non-growing *Hyp* mice. Nevertheless, the findings in the current study led us to hypothesize that hypertension in aged *Hyp* mice may be caused by increased RAAS signaling alone or in combination with FGF23-mediated increased sodium absorption, leading to a combination of volume overload and increased peripheral vascular resistance. Aldosterone, similar to FGF23, enhances sodium and water reabsorption in the distal nephron, indirectly increasing blood pressure [24,25,26]. Furthermore, it was shown previously that Ang II administration to *Hyp* mice resulted in additive effects on blood pressure [21]. We used short term (5-day) treatment with the mineralocorticoid receptor blocker canrenone and the angiotensin II receptor type 1 antagonist losartan as tools to dissect the renal and cardiovascular effects of aldosterone and angiotensin II. However, both drugs lowered blood pressure and decreased peripheral resistance in aged *Hyp* mice. Therefore, it is likely that a combination of volume effects and increased peripheral vascular resistance is involved in the hypertension-promoting effects of RAAS stimulation in aged *Hyp* mice. Interestingly, these drugs had little effect in WT mice, further corroborating that augmented RAAS signaling contributes to the development of hypertension in *Hyp* mice.

Although our study has provided novel insights into the pathogenesis of hypertension in *Hyp* mice, a key question in this context remains unanswered: what is driving the stimulation of RAAS in aged *Hyp* mice? It has been suggested that FGF23 hypersecretion may stimulate the RAAS through the suppression of vitamin D hormone production in CKD [27,28]. However, whether RAAS is regulated by vitamin D remains a controversial issue [29], and it is currently not known if this proposed mechanism may also be relevant for *Hyp* mice. It is clear that further experimentation is required to define the molecular basis of the crosstalk between FGF23 and the RAAS. This is not only relevant for XLH patients, but also has much broader implications for diseases characterized by a combination of chronic FGF23 hypersecretion and increased RAAS signaling, such as CKD [30].

## Figures and Tables

**Figure 1 biomedicines-10-01691-f001:**
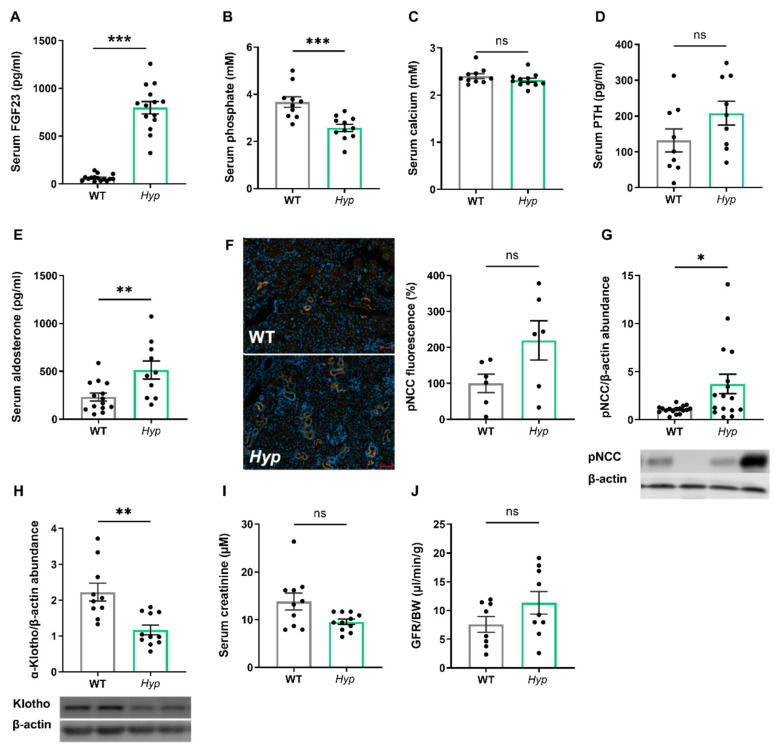
**Aged *Hyp* mice show a stimulation of the RAAS and a down-regulation of renal αKlotho protein abundance, but have normal kidney function.** (**A**) Serum intact FGF23 levels (n = 14–15), (**B**,**C**) serum phosphate and calcium concentrations (n = 10–11), and (**D**,**E**) serum intact PTH and aldosterone levels (n = 9–14) in 14-month-old male wildtype (WT) and *Hyp* mice. (**F**) Left: representative images of immunofluorescent staining of kidney paraffin sections with an anti-pNCC antibody (original magnification 400×). Right: quantification of anti-pNCC immunofluorescence in the kidney in 14-month-old male WT and *Hyp* mice (n = 6). Western blot quantification of (**G**) pNCC and (**H**) α-Klotho from kidney cortex homogenates (n = 6–11), as well as (**I**) serum creatinine levels (n = 10–11) and (**J**) glomerular filtration rate per gram body weight as measured by renal creatinine clearance (n = 8–9) in 14-month-old male *Hyp* mice and WT controls. Bars in (**A**–**J**) represent mean ± SEM for WT and *Hyp* mice. * *p* < 0.05, ** *p* < 0.01, *** *p* < 0.001 by Student’s *t*-test. ns, not significant.

**Figure 2 biomedicines-10-01691-f002:**
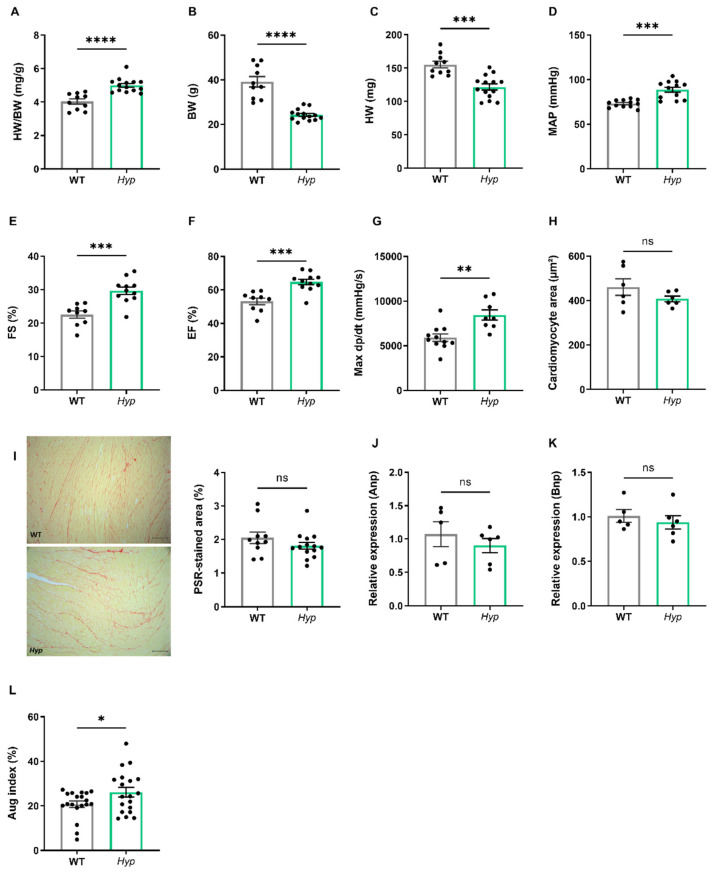
***Hyp* mice are hypertensive but do not develop LV hypertrophy.** (**A**) Heart/body weight ratio (HW/BW), (**B**) body weight (BW), and (**C**) heart weight (HW) (n = 4–14) in 14-month-old male WT and *Hyp* mice. (**D**) Mean arterial pressure measured by arterial catheterization (n = 12), (**E**,**F**) fractional shortening (FS) and ejection fraction (EF) measured by echocardiography (n = 9–11), and (**G**) LV contractility (Max dP/dt) measured by intracardiac catheterization (n = 12) in 14-month-old male WT and *Hyp* mice. (**H**) Quantification of mean cardiomyocyte size after FITC-WGA staining (n = 6), (**I**) left, representative images of collagen staining using picrosirius red (PSR) in cardiac paraffin sections (bar = 100 μm), right, quantification of fibrosis as measured by PSR-stained area (n = 9–12), (**J**,**K**) relative mRNA expression of markers of hypertrophy, (**J**) atrial natriuretic peptide (Anp) and (**K**) brain natriuretic peptide (Bnp) (n = 5), and (**L**) augmentation index (AI) measured by pulse wave analysis (n = 19) in 14-month-old male WT and *Hyp* mice. Bars in (**A**–**L**) represent mean ± SEM. * *p* < 0.05, ** *p* < 0.01, *** *p* < 0.001, **** *p* < 0.0001 by Student’s *t*-test. ns, not significant.

**Figure 3 biomedicines-10-01691-f003:**
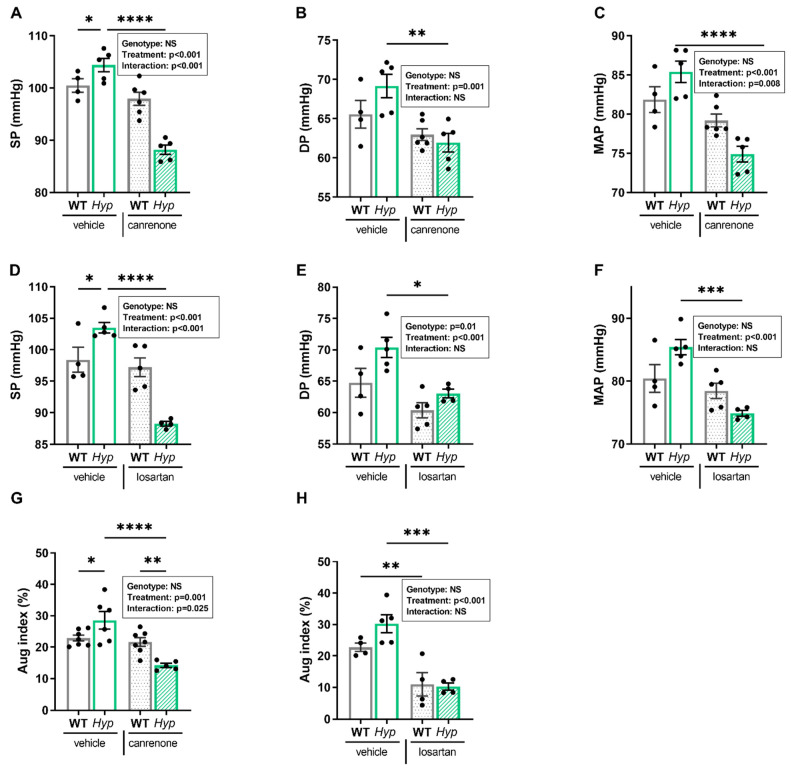
**Inhibition of RAAS signaling normalizes blood pressure in *Hyp* mice.** (**A**,**D**) Systolic (SP), (**B**,**E**) diastolic (DP), and (**C**,**F**) mean arterial blood pressure (MAP), as well as (**G**,**H**) augmentation index (AI) in 12–14-month-old male WT and *Hyp* mice treated over 5 days with 30 mg/kg canrenone (s.c. once daily) or 30 mg/kg losartan (gavage twice daily) (n = 4–8). Bars represent mean ± SEM. * *p* < 0.05, ** *p* < 0.01, *** *p* < 0.001, **** *p* < 0.0001 vs. vehicle by one-way ANOVA followed by Student–Newman–Keuls post-hoc test. Insets show results of two-way ANOVA.

## Data Availability

All data generated or analyzed in this study are included in this article.

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
