# Peer review of "Activation of RAAS Signaling Contributes to Hypertension in Aged Hyp Mice"

_biomedicines, 2022, doi:10.3390/biomedicines10071691_

Round 1

Reviewer 1 Report

This manuscript is aimed to investigate the long-term cardiovascular sequelae of elevated circulating levels of FGF23 in 14-month-old Hyp mice. The authors found that aged Hyp mice didn’t exhibit LV hypertrophy, but were characterized by increased aldosterone and hypertension, which could be abolished by inhibitors of RAAS system. However, it’s obscure to determine the duration as “long-term”, as 9-month-old Hyp mice were studied in previous studies. In addition, it is an observational study without any experiments to elucidate the underlying mechanisms, the novelty of this study is greatly dampened. There are quite a lot of concerns still needed to be addressed before further consideration by the journal.

1.     How did the authors determine the dosage of losartan and canrenone in the animal experiment? 

2.     Why was the serum level of FGF23 only detected in 4 Hyp mice, while serum phosphate, calcium, PTH and aldosterone were examined in more than 9 Hyp mice? The ratio of GFR to body weight was also detected in 5 Hyp mice. 

3.     Why only male mice (WT and Hyp mice) were involved in the experiment? Is there any difference in serum FGF23, phosphate, calcium, PTH and aldosterone between male and female Hyp mice?

4.     It’s very interesting that 3-month-old Hyp mice had lower serum aldosterone levels, while aged Hyp mice were characterized by higher serum aldosterone. Please discuss the possible reasons that might explain the opposite changes of aldosterone between the earlier and advanced stage of Hyp mice and add the discussion into the main text where appropriate. It would also be more interesting to explore the molecular mechanism underlying the difference in aldosterone at earlier and advanced stage of Hyp mice.

5.     It was mentioned that Hyp mice of up to nine months of age didn’t develop LV hypertrophy. The 14 month-old Hyp mice was then used as a model of long-term FGF23 excess. I was wondering whether FGF23 was already higher in 9 month-old Hyp mice. Did the authors detect FGF23 levels at different time point (3 month, 6 month, 9 month, 12 month etc.) in Hyp mice? Since there are lines of studies focusing on the relationship between FGF23 and RAAS and this study is observational without any experiments to elucidate the underlying mechanisms, the novelty of this study is greatly dampened. It would be highly recommended checking the transcriptomic or metabolic reprogramming in Hyp mice, thus shedding some light on the molecular mechanism related to the development of hypertension in this disease setting.

6.     The resolution of representative images in Fig 2i was low, please replace them with images of higher resolution.

Reviewer 2 Report

This is a study testing the hypothesis that a stimulation of the renin-angiotensin-aldosterone system (RAAS) contributes to hypertension in aged Hyp mice.  Angiotensin receptor blocker losartan (30 mg/kg twice daily) or mineralocorticoid receptor antagonist canrenone (30 mg/kg once daily) were given to aged Hyp and WT mice over 5 days. The results showed that both drugs had minor effects on blood pressure in WT mice, but reduced blood pressure and peripheral vascular resistance in Hyp mice, suggesting that a stimulation of the RAAS contributes to hypertension in aged Hyp mice. 

The methodology was standard and not new with echo of the ventricle and catheterization for intra-arterial pressure.

The results were intriguing and showed clear involvement of the RAAS which was not new. This study contributed more to the understanding of the mechanism of HTN   

Author Response

Response to reviewers, biomedicines-1768376

We would like to thank the reviewers for their constructive critique which has significantly strengthened this manuscript.

Reviewer 2:

The methodology was standard and not new with echo of the ventricle and catheterization for intra-arterial pressure.

The results were intriguing and showed clear involvement of the RAAS which was not new. This study contributed more to the understanding of the mechanism of HTN.

Authors: We agree that there is already data discussing the interaction of FGF23 and RAAS in mice, but the cardiovascular effects of RAAS antagonists in hypertensive, aged Hyp mice have not been investigated before. Our study has clearly provided novel insights into the pathogenesis of hypertension in Hyp mice.

Round 2

Reviewer 1 Report

All the revisions made by the authors are noted and acknowledged. I congratulate the authors for making the necessary revisions or providing explanations and clarifications to my previous concerns. Therefore, I have no further questions at the moment and would like to recommend accepting the current version.